**Protocol**

# Core Outcome Sets in Miscarriage Trials (COSMisT) study: a study protocol

Paul Smith,[1] Natalie Cooper,[2] Rima Dhillon-Smith,[1] Emily O'Toole,[3] T Justin Clark,[1] Arri Coomarasamy[1]

[1]Academic Department, Institute of Metabolism and Systems Research, University of Birmingham, Birmingham Women's Hospital, Birmingham, West Midlands, UK
[2]Women's Health Research Unit, Barts and The London School of Medicine, Queen Mary University of London, London and Barts Health NHS Trust, London, UK
[3]Women's Voices Involvement Panel, Royal College of Obstetricians and Gynaecologists, London, UK

**Correspondence to**
Dr Paul Smith;
paul.smith@doctors.org.uk

## ABSTRACT

**Introduction** 'Core outcome sets' are an agreed, standardised set of outcomes based on what key stakeholders (clinicians, patients, their partners, researchers, service developers, funding organisations and so on) consider the important outcomes in the management or prevention of a condition. This paper describes the rationale and design for the development of Core Outcome Sets for Miscarriage Trials.

**Methods and analysis** Systematic reviews, interviews and focus groups with patients and their partners will be conducted to identify potential core outcomes that will be introduced into a modified Delphi survey. To ensure all key stakeholders are included, patients, partners, clinicians, charities and researchers will be invited to take part in the modified Delphi survey. There will be three rounds of scoring and rescoring during the Delphi survey to reach consensus regarding outcomes to be included in the core set, which will be subsequently refined through face-to-face consensus discussions.

**Ethics and dissemination** The use of core outcome sets allows results from different studies to be compared and combined, thereby reducing inconsistency and aiding interpretation of study findings. It also means research is more likely to report relevant outcomes and so can reduce reporting bias. Understanding which outcomes are important to patients has the potential to act as a driver to improve both the quality and cost-effectiveness of miscarriage services.

## INTRODUCTION

Miscarriage is defined as the premature loss of a pregnancy up to 23 weeks of gestation or weighing less than 500 g.[1] An estimated 140 000 women per year suffer from miscarriage, costing the National Health Service over £350 million/year. The clinical symptoms of miscarriage are vaginal bleeding usually associated with abdominal pain. It is thought that 15%–20% of all pregnancies will end in miscarriage,[2] and this can have psychological and medical consequences on both women and their partners.[3–6]

Although miscarriage is the biggest cause of pregnancy loss in the UK, it remains poorly understood and there is a great deal of research focused on improving treatment and services for this condition. Clinical trials, systematic reviews and guidelines have compared various interventions for the prevention and management of miscarriage. However, studies on miscarriage often do not address the same outcomes, making it difficult to draw conclusions when the evidence is synthesised. Thus, a standardised collection of relevant outcomes is needed to aid the interpretation of study findings. This can be achieved through the development of a 'core outcome set'. Core outcome sets are an agreed, standardised set of outcomes based on what key stakeholders (clinicians, patients, their partners, researchers, service developers, funding organisations and so on) consider the important outcomes in the management or prevention of a condition.[7]

Core outcomes are disease-specific and take into account both the potential benefits and harms of interventions. The use of core outcome sets reduces inconsistencies, allowing results from different studies to be compared and combined. It also means research is more likely to report relevant outcomes and reporting bias may be minimised. This is because researchers are expected to report on all the core outcomes or state explicitly why particular outcomes are not reported.

The measurement of patient-reported outcome measures (PROMs) in clinical trials has increased significantly in the last 20 years.[8] The importance of involving patients in trial research is now well acknowledged. The Core Outcome Measures in Effectiveness Trials (COMET) initiative advocates the involvement of patients and lay members of the public in decisions about which endpoints should be included in core outcome sets within healthcare. Core outcome sets are now being developed for a number of clinical conditions, and in the UK the National Institute for Health Research, Health Technology Assessment and the Cochrane Collaboration advocate their use.[7 9] An increased understanding about which outcomes are important to patients has the potential to act

as a driver to improve both the quality and cost-effectiveness of miscarriage services.

This paper describes a proposed study design for the development of Core Outcome Sets in Miscarriage Trials (COSMisT).

## Study objectives

1. To systematically review and collate outcomes already used for prevention and management of miscarriage.
2. To establish outcomes that matter most to patients using qualitative methods.
3. To get all key stakeholders to agree on a core outcome set that will be used in future research on the management and prevention of miscarriage.
4. To disseminate core outcome sets for miscarriage.

## METHODS

The core outcome set in miscarriage prevention and treatment was prospectively registered with the COMET initiative under registration numbers 679, 815 and 816 (http://www.comet-initiative.org). Methodology, endorsed by COMET, which engages all key stakeholders (clinicians, patients, charities and researchers), will be used to develop the core outcome set.[10] In order to identify potential core outcomes, systematic literature reviews and focus groups with stakeholders will be conducted. Following this, a Delphi survey will be implemented to reach a consensus on which outcomes should be included within the core set; subsequently this final core set will be refined through face-to-face discussions. A description of each stage is detailed below:

### Identifying possible outcomes
#### Systematic review

A comprehensive systematic review will be performed to identify the outcomes used in studies evaluating expectant management, medical management, surgical management and prevention of early miscarriage. The systematic review has been registered on PROSPERO International Prospective Register of Systematic Reviews (14/3/16, ID: PROSPERO 2016:CRD42016036349). Literature searches will be conducted on the following databases: MEDLINE (from 1946 to date), EMBASE (from 1980 to date), Cumulative Index to Nursing and Allied Health Literature (from 1981 to date) and the Cochrane Central Register of Controlled Trial and clinical trial registries. A search strategy using both medical subject headings and keywords in the title and/or abstract and subject headings will be used. Studies that will be included will be any randomised controlled trial looking at the prevention or treatment of early miscarriage. For the purpose of the systematic review, we will define early miscarriage as pregnancy loss in the first trimester. The research question has been limited to first trimester losses as the prevention strategies and clinical management of second trimester losses can widely differ from that of first trimester. Subsequently trying to merge these would result in complex analysis and difficulty

extrapolating and comparing results. However, there are similarities in the causes and management of pregnancy losses between the two trimesters, and so by using studies that include first trimester losses these will be captured. We will restrict the search to randomised controlled studies and exclude quasi-randomised, non-randomised trials, observational studies, diagnostic studies, case-series, case reports and surveys. No language restrictions will be applied to the search.

The study selection will be a two-stage process conducted by two independent reviewers. The first stage will entail screening titles and abstracts against the inclusion criteria to identify relevant papers. The second stage will be to scan the manuscripts of the citations that fulfilled the predefined selection criteria. If there is a difference of opinion, a third-party arbitrator will be involved to reach consensus. Data extraction from studies fulfilling the inclusion criteria will be done in duplicate with discussion and consensus used to overcome disagreements. Data will be extracted for the type of miscarriage study (medical management, surgical management, expectant management or prevention of early miscarriage), study aims and outcome information (primary, secondary, outcome definitions and outcomes measures) using a customised data extraction form. The outcomes identified through the systematic review will be compiled and added to the list to be considered for the Delphi survey.

### Qualitative work

It has been increasingly realised that the development of patient-centred outcomes is an important component of core outcome sets. Patient-centred outcomes have traditionally been left out of clinical research. To ensure that these are not missed, we will invite patients and their partners to take part in focus groups or interviews to identify potential core outcomes and also explore why outcomes are important to them.

Potential participants will be identified and invited to participate through screening of health records and by open invitation using posters and information leaflets. This will be done in early pregnancy units, inpatient wards and with the help of support groups and online communities. Inclusion criteria will include women who have suffered from an early miscarriage or partners of women who have suffered an early miscarriage. There will be no restrictions on how the women got pregnant (ie, assisted conception pregnancies or spontaneous), the number of miscarriages or how long ago the miscarriage was.

Research using qualitative interviews to acquire PROMs suggests that a minimum of 20 patients are required before data saturation tends to be reached.[11 12] We will aim to recruit a group of between 8 and 15 participants per workshop as larger groups would be hard to facilitate.[13] We will aim to have at least two focus groups and would look to recruit between 20 and 60 people in total. We will have a sampling strategy of maximum variation with the aim to have a range of ages and miscarriage experiences.

To begin the focus group an introduction will be given by a facilitator who will explain the purpose of the study and give the working principles of the group. The focus groups will then be asked to cast their mind back to when they experienced miscarriage and consider what was important to them and write it down on a piece of paper. This could have been anything that is important to the woman or her partner from physical symptoms to worries or effects on daily activities. These points will then be read out and discussed among the group anonymously along with any other outcomes that are brought up during the discussion. The main aim will be to generate patient-centred outcomes and to find out why they have been chosen and how important they are to the group. To allow for exploration of underlying themes, the group discussions will be recorded. Participants will be asked to prioritise the outcomes in order of what is most important to them; this is so that at a later stage (following the Delphi survey) we can see which outcomes were originally identified as most important to the patients and their partners.

Semistructured interviews may be necessary because of a limited number of available participants or difficulty recruiting participants to talk about a sensitive subject. The aim would be to achieve thematic saturation, and a pragmatic approach will be adopted in order to reach this. At least 10 participants will be interviewed, checking for thematic saturation. All interviews conducted will be audio-recorded, transcribed in full and subsequently coded. To increase the validity of the findings, any deviant (ie, discordant) themes will be given special consideration.

## Outcome list

To ensure there is no replication of outcomes, the core outcomes implementation group will review the list of outcomes collated from the literature, group sessions and interviews prior to commencing the consensus survey.

## Survey of stakeholders using Delphi methodology
### The Delphi panel

To ensure the quality of the core outcome set, a range of expertise within the panel will be important. With this in mind we will seek to recruit members of the panel from the following key groups:

1. Patients and/or partners: the aim will be to maximise the variation in miscarriage experience and range of ages of women and/or their partners.
2. Researchers: academics who have a particular interest in performing trials in early pregnancy and miscarriage.
3. Physicians: clinicians who have an interest in early pregnancy.
4. Nurses and allied healthcare professionals.
5. Charities or patient support groups.

We will try to recruit as comprehensive a panel as possible, ensuring we get participants from all the key stakeholder groups. To ensure that the core outcomes set is useful outside the UK, we will try to enrol international participants for the Delphi survey. There will be a number

of questions at the start of the survey to ensure that participants are eligible to take part.

### Delphi survey

There is limited empirical evidence on the number of participants required for a Delphi survey.[13] We will adopt a pragmatic approach and ensure that participants are recruited from all stakeholder groups and in manageable numbers to allow for the continuation of the Delphi survey through all three rounds. Participants will be asked to complete either an online Delphi survey or a paper version depending on their preference. The Delphi survey will be made up of three rounds:

### Round 1

The proposed outcomes collated from the literature reviews and participant group sessions will be compiled in alphabetical order. Using the Grading of Recommendations, Assessment, Development and Evaluations scale,[14] participants will be asked to score each outcome to reflect 'how important' they feel it is. The scale will range from 1 to 9 and will be categorised as follows: 1–3 'not important'; 4–6 'important but not critical'; and 7–9 'critical'. Following completion of the survey, participants will be asked to suggest any other outcomes that they think are relevant or important which have not been covered by the survey.

### Round 2

A summary of the distribution of scores for the whole group for each outcome during round 1 will be created. This summary of scores will then be sent to the participants and they will be required to again score each outcome in order of importance in view of the 'whole group' scores.

### Round 3

Once again, the participants will be sent a summary of the distribution of scores. They will once more be required to rescore the outcomes as previously described and also to state whether they think each outcome should be included in the final core outcome set. The aim of repeating the process over three rounds is to encourage convergence of ideas. The criteria for determining which outcomes should be included in the core outcome set (consensus in) will be that >70% participants score the outcome as critical (7–9), while <15% score it as not important (1–3).[15] The reverse will apply for outcomes considered 'consensus out'. For any outcomes where there is no consensus reached, there will be further evaluation in consensus meetings to decide how best to deal with these. There may be outcomes for which a consensus is not reached (no consensus), which will require further evaluation in consensus meetings.

### Consensus meeting

Once the systematic review, qualitative research and Delphi process are completed, a consensus meeting will be arranged between key stakeholders and the research management team. The main aim will be to discuss the best way to disseminate the results, but if consensus has not been

reached on all the outcomes the group will discuss these. During the meeting, consensus outcomes from the Delphi survey will be reviewed; outcomes for which consensus was not reached will be discussed (bearing in mind results from service user workshops as service users will not join these meetings) and the core outcome set will be finalised. For any outcomes where a consensus is not reached or there remains disagreement, these will be evaluated against a prespecified criteria using the RAND disagreement.[16]

## Ethics and dissemination

### Ethics

The conduct of the research will be done in accordance with the guidance of good clinical practice and in keeping with local regulations.

### Dissemination

A core outcome set will only make an impact if it is consistently implemented in trials, and so to ensure this happens efforts will be made to enthusiastically engage with regulators, trialists, journals and funders to ensure that core outcomes in miscarriage are used, not just in the UK but on a global scale. A manuscript will be prepared with the primary results of the study and submitted to peer-reviewed journals. It is anticipated that this research will be presented at national and international conferences to further disseminate this work.

## DISCUSSION

The COSMiST study will be the first core outcome set in miscarriage and will provide a standardised set of outcome measures. It will have the potential to influence and improve the clinical experience of the patients by developing patient-centred outcomes that can be incorporated into clinical practice and make the results of trials more clinically meaningful. More specifically, core outcome sets for miscarriage will ensure that trials add to the existing evidence for miscarriage by improving the reporting and conduct of the clinical trials. It will harmonise outcomes and allow results from different trials to be combined and compared in systematic reviews, meta-analysis and clinical guidelines. In this way, the more effective interventions can be identified and subsequently implemented to improve patient experience and outcomes.

A common misconception of core outcome sets is that they comprise the *only* outcomes that should be reported in trials within that specific area. This contention is erroneous because the intent of generating core outcome sets is to provide a *minimum* list of outcomes that should be reported for a trial in that subject. The framework provided by the core outcome set can be added to or trimmed depending on the needs of the research. It remains incumbent on the researcher to explicitly state why a core outcome has not been included or why other outcomes have been collected.

**Contributors** PS and NC conceived the idea, developed the research question and study methodology, and contributed meaningfully to the drafting and editing of the final manuscript. RD-S, EO, TJC and AC aided in developing the research question and study methodology, and contributed meaningfully to the drafting and editing of the final manuscript.

**Funding** The research costs of the trial are funded as part of a postdoctoral fellowship award from the National Institute for Health Research (NIHR), reference: PDF-2015-08-099.

**Disclaimer** This article/paper/report presents independent research funded by the National Institute for Health Research (NIHR) (and Health Education England if applicable). The views expressed are those of the author(s) and not necessarily those of the NHS, the NIHR or the Department of Health.

**Competing interests** None declared.

**Ethics approval** Since the systematic review consists of collecting and reviewing publicly available data, ethical approval is not required. To identify potential core outcomes through focus groups or interviews, the trial has been awarded ethical approval by South Birmingham Research Ethics Committee, IRAS project ID: 1996321 (protocol version 1.2, dated 7 July 2016).

**Provenance and peer review** Not commissioned; externally peer reviewed.

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
