## [Reviewer comments · BMJ Open]

ARTICLE DETAILS

TITLE (PROVISIONAL)	Core Outcome Sets in Miscarriage Trials (COSMisT) Study: a study protocol
AUTHORS	Smith, Paul; Cooper, Natalie; Dhillon-Smith, Rima; O'Toole, Emily; Clark, T; Coomarasamy, Arri

VERSION 1 – REVIEW

REVIEWER	Merel MJ van den Berg Centre of Reproductive Medicine, Academic Medical Centre, Amsterdam, The Netherlands
REVIEW RETURNED	31-Jul-2017

GENERAL COMMENTS	Summary. This article describes a study protocol for the development of a standardized set of outcomes for miscarriage trials. The Delphi method will be used to identify the set of outcomes. The topic of this article is relevant since miscarriage is a common and complex clinical problem. A lot of research is done in the field of treatment of miscarriage. A standardized set of outcomes will help to compare and interpret the different studies in this field. Overall opinion of the manuscript. The study protocol is well described. The focus of research has been shifted to patient-centered care. Thereby, it's really good that they will involve patients in the progress of the development of the set. I hope it is possible to include the necessary amount of patients in the focusgroups, because it's a sensitive topic. General comments. 1. Numbering of the lines is not continuously through all pages.2. For nomenclature I suggest the authors to read Kolte et al HR 2015 about suggested nomenclature in early pregnancy research. Specific comments. Page 7: You will define early miscarriage as pregnancy loss in the first trimester. However, in the introduction you defined miscarriage as the premature loss of a pregnancy up to 23 weeks of gestation. Why do you choose to deviate from this earlier mentioned definition? Please motivate this decision in the manuscript. Page 7: The inclusion criteria used for the systematic review are not described clearly in the method section. Please include the specific criteria in the manuscript.
--

	Page 7: You mention that data will be extracted for the type of miscarriage study, i.e. medical management, surgical management or prevention of early miscarriage. However, I miss the option of expectant management. Can you motivate why you don't include expectant treatment? I think you also should include expectant management. Page 8: The inclusion criteria for the focus groups are described here. Can you be more specific: is there a criterion for the amount of miscarriages? When should the patients have suffered from a miscarriage, i.e. how long ago? Do you include couples only, or are women without their partners also welcome? Are there criteria for the way that couples got pregnant, i.e. IVF pregnancies or only spontaneous? Page 9: For the identification of the key group patients and partners you mention that you want to include in particular patients who suffered from recurrent miscarriage. You don't mention this criterion before. Can you motivate this decision? Do you think there will be a difference between patients who suffered from miscarriage and recurrent miscarriage? Page 10: I'm used to include a top 5 after every round in the Delphi method. Why don't you use the top 5?
--	---

REVIEWER	David Cahill University of Bristol UK None I am aware of
REVIEW RETURNED	31-Jul-2017
GENERAL COMMENTS	Generally this paper is over long and in that, it fails to precisely define the methodology and the purpose of the paper. A bit of fierce editing could focus that more effectively

VERSION 1 – AUTHOR RESPONSE

Reviewer: 1

SUMMARY OF THE CONTENTS:

This article describes a study protocol for the development of a standardized set of outcomes for miscarriage trials. The Delphi method will be used to identify the set of outcomes. The topic of this article is relevant since miscarriage is a common and complex clinical problem. A lot of research is done in the field of treatment of miscarriage. A standardized set of outcomes will help to compare and interpret the different studies in this field.

SPECIFIC COMMENTS

Comment: Page 7: You will define early miscarriage as pregnancy loss in the first trimester. However, in the introduction you defined miscarriage as the premature loss of a pregnancy up to 23 weeks of gestation. Why do you choose to deviate from this earlier mentioned definition? Please motivate this decision in the manuscript.

Response: We thank the reviewer for this comment and admit this was an area debated in the creation of the protocol initially. It has been agreed amongst the study team that the prevention strategies and clinical management of 2nd trimester losses can differ widely from that of 1st trimester losses; for example the use of cerclage for cervical incompetence and medical disorders only relevant to 2nd trimester i.e. Pre-eclampsia. To combine 1st and 2nd trimester losses would be not be appropriate and would result in a very complex study protocol, and we felt looking at 2nd trimester losses would almost warrant a separate study of its own given the volume of existing literature. For the above reasons it was decided to limit to 1st trimester only. However we will ensure any conditions, which could affect both 1st and 2nd trimester losses, will be captured.

The following sentences have been added to the methods section:

“The research question has been limited to first trimester losses as the prevention strategies and clinical management of second trimester losses can widely differ from that of first trimester. Subsequently trying to merge these would result in complex analysis and difficulty extrapolating and comparing results. However, there are similarities in the causes and management of pregnancy losses between the two trimesters, and so by using studies that include first trimester losses these will be captured“.

Comment: Page 7: The inclusion criteria used for the systematic review are not described clearly in the method section. Please include the specific criteria in the manuscript.

Response: We have modified the methods section to describe the inclusion and exclusion criteria more clearly:

“Studies that will be included will be any randomized controlled trial looking at the prevention or treatment of early miscarriage. For the purpose of the systematic review, we will define early miscarriage as pregnancy loss in the first trimester. The research question has been limited to first trimester losses as the prevention strategies and clinical management of second-trimester losses can widely differ from that of first-trimester. Subsequently trying to merge these would result in complex analysis and difficulty extrapolating and comparing results. However, there are similarities in the causes and management of pregnancy losses between the two trimesters, and so by using studies that include first trimester losses, these will be captured. We will restrict the search to randomised controlled studies and exclude quasi-randomised, non-randomised trials, observational studies, diagnostic studies, case-series, case reports and surveys. No language restrictions will be applied to the search.”

Comment: Page 7: You mention that data will be extracted for the type of miscarriage study, i.e. medical management, surgical management or prevention of early miscarriage. However, I miss the option of expectant management. Can you motivate why you don't include expectant treatment? I think you also should include expectant management.

Response: We agree with the reviewer that expectant management is important as stated in the opening sentence of the paragraph “A comprehensive systematic review will be performed to identify the outcomes used in studies evaluating expectant management, medical management, surgical management and prevention of early miscarriage.” This has now been further clarified in the manuscript in the subsection relating to data extraction: “Data will be extracted for the type of miscarriage study (medical management, surgical management, expectant management or prevention of early miscarriage)”.

Comment: Page 8: The inclusion criteria for the focus groups are described here. Can you be more specific: is there a criterion for a number of miscarriages? When should the patients have suffered from a miscarriage, i.e. how long ago? Do you include couples only, or are women without their partners also welcome? Are there criteria for the way that couples got pregnant, i.e. IVF pregnancies or only spontaneous?

Response: We thank the reviewer for highlighting the lack of clarity here. We have now clarified the inclusion and exclusion criteria: "Inclusion criteria will include women who have suffered from an early miscarriage or partners of women who have suffered an early miscarriage. There will be no restrictions on pregnancy conception (i.e. IVF or spontaneous), the number of miscarriages or date of miscarriage".

Comment: Page 9: For the identification of the key group patients and partners you mention that you want to include in particular patients who suffered from recurrent miscarriage. You don't mention this criterion before. Can you motivate this decision? Do you think there will be a difference between patients who suffered from miscarriage and recurrent miscarriage?

Response: We thank the reviewer for this feedback. We are aiming to create a diverse group of people who have had different experiences and potentially different views. We do not know if women with recurrent miscarriage will feel differently, but we would like women with a variety of experiences to secure these potential different views. We have now changed the sentence to "Patients and/or partners: the aim will be to maximize the variation in miscarriage experience and range of ages of women and/or their partners".

Comment: Page 10: I'm used to include a top 5 after every round in the Delphi method. Why don't you use the top 5?

Response: We thank the reviewer for suggesting an alternative methodology however this is the methodology we are familiar with and we do not want to limit the output of this work to a pre-specified number of outcomes.

Reviewer: 2

Comment: Generally this paper is over long and in that, it fails to precisely define the methodology and the purpose of the paper. A bit of fierce editing could focus that more effectively

Response: We thank the reviewer for their comments. Without the reviewer being more precise it is hard to make specific changes. We have however made changes to the methodology as advised by Reviewer 1 which we feel focuses the aim of the study and clarifies it's purpose further. We hope that the changes made are sufficient for the paper to now be accepted for publication. We look forward to your response.